# Comprehensive Review on Bimolecular Fluorescence Complementation and Its Application in Deciphering Protein–Protein Interactions in Cell Signaling Pathways

**DOI:** 10.3390/biom14070859

**Published:** 2024-07-17

**Authors:** Houming Ren, Qingshan Ou, Qian Pu, Yuqi Lou, Xiaolin Yang, Yujiao Han, Shiping Liu

**Affiliations:** State Key Laboratory of Resource Insects, Southwest University, Chongqing 400716, China; rhm7833097@email.swu.edu.cn (H.R.); oqs123033@email.swu.edu.cn (Q.O.); pq7426@email.swu.edu.cn (Q.P.); quqi617@email.swu.edu.cn (Y.L.); yangxiaolin@email.swu.edu.cn (X.Y.); hyj1998@email.swu.edu.cn (Y.H.)

**Keywords:** bimolecular fluorescence complementation (BiFC), protein–protein interactions (PPIs), cell signaling pathway

## Abstract

Signaling pathways are responsible for transmitting information between cells and regulating cell growth, differentiation, and death. Proteins in cells form complexes by interacting with each other through specific structural domains, playing a crucial role in various biological functions and cell signaling pathways. Protein–protein interactions (PPIs) within cell signaling pathways are essential for signal transmission and regulation. The spatiotemporal features of PPIs in signaling pathways are crucial for comprehending the regulatory mechanisms of signal transduction. Bimolecular fluorescence complementation (BiFC) is one kind of imaging tool for the direct visualization of PPIs in living cells and has been widely utilized to uncover novel PPIs in various organisms. BiFC demonstrates significant potential for application in various areas of biological research, drug development, disease diagnosis and treatment, and other related fields. This review systematically summarizes and analyzes the technical advancement of BiFC and its utilization in elucidating PPIs within established cell signaling pathways, including TOR, PI3K/Akt, Wnt/β-catenin, NF-κB, and MAPK. Additionally, it explores the application of this technology in revealing PPIs within the plant hormone signaling pathways of ethylene, auxin, Gibberellin, and abscisic acid. Using BiFC in conjunction with CRISPR-Cas9, live-cell imaging, and ultra-high-resolution microscopy will enhance our comprehension of PPIs in cell signaling pathways.

## 1. Introduction

Cellular signaling pathways are intricate biochemical reactions and play a crucial role in facilitating information transmission in living organisms, enabling cells to promptly and accurately respond to external stimuli [1,2,3,4]. The transmission of signals in biological cells is primarily facilitated through molecular interactions, particularly protein–protein interactions (PPI) [5,6,7,8]. The real-time observation of PPI dynamics, alterations, and spatial-temporal distribution is crucial for elucidating the regulatory interplay among various genes. Conventional biochemical techniques utilized for the examination of PPI, such as co-immunoprecipitation (Co-IP) and immunofluorescence (IF), are constrained in their ability to offer insights into the transient and dynamic nature of interactions as well as to provide accurate and detailed visualization. In light of the advancements in molecular biology and biotechnology, researchers have been actively investigating novel approaches to elucidate these intricate signaling pathways. Notably, bimolecular fluorescence complementarity (BiFC) has emerged as a valuable tool to explore protein interactions, based on the division and recombination of fluorescent proteins, which are usually divided into an N-terminal fragment and a C-terminal fragment [9,10]. When two target proteins interact with each other, the N-terminal and C-terminal fragments of the fluorescent protein will approach each other and recombine to restore the original structure and properties, thus generating fluorescence signals to reflect the interaction of the target proteins, offering substantial assistance in the examination of cellular signaling pathways [10,11,12] (Figure 1A). BiFC has significantly contributed to the investigation of cellular signaling pathways by enabling the examination of interactions between signaling proteins and their downstream effector proteins as well as the analysis of signal transmission within cells [13,14,15]. This visualization technique aids in comprehending the composition and structure of signaling pathways as well as in analyzing the dynamic alterations of signaling pathways in various physiological and pathological states [16,17,18,19,20]. Its utility extends to the study of fundamental biological processes including cell signal transduction, gene expression regulation, and protein complex formation [12,21,22]. By conducting comprehensive investigations into these processes, valuable insights into the underlying principles and mechanisms governing life activities can be obtained. Collectively, BiFC allows for the in situ detection of PPIs, real-time monitoring of cellular activity, and examination of subcellular localization, thus offering a distinct viewpoint for uncovering critical regulatory relationships and signaling pathways within gene regulatory networks.

Since the emergence of BiFC in the 1990s, there have been several notable advancements in this technology. One such advancement is the development of split GFP variants and other fluorescent proteins with improved brightness and photostability, allowing for more sensitive and quantitative detection of PPIs [10,15,23,24,25]. The simultaneous expression of two or more fluorescent proteins in a multicolor bimolecular fluorescence complementation (mcBiFC) system facilitates the concurrent monitoring of multiple protein interactions and allows for the comparison of their interaction efficiencies [26,27,28] (Figure 1B). Another advancement is the use of BiFC in high-throughput screens, enabling the rapid identification of PPIs on a genome-wide scale [29,30,31,32]. BiFC has also been developed to visualize the interaction between RNA and proteins, as evidenced by trimeric fluorescence complementation (TriFC) systems [33] (Figure 1C). TriFC is optimized by utilizing mNeptune’s good optical properties to image RNA–protein interactions in living animals [34]. The integration of dCas13a with SunTag has been utilized to enhance the recruitment of Venus fragments, thereby enabling the real-time visualization of endogenous specific mRNA in cells through the targeted binding of gRNA with cas13a [35] (Figure 1D). The fusion of BiFC with transcription activator-like effectors (TALEs) enables precise labeling and visualization of genomic loci through the specific interaction of TALEs with target DNA sequences [36] (Figure 1E).

BiFC exhibits significant promise in the fields of disease diagnosis and drug screening. By constructing large-scale PPI networks, researchers can screen for key signaling pathways associated with specific diseases and further investigate potential drug targets in these pathways [37,38,39,40,41,42]. BiFC is a valuable tool for screening potential drug molecules and investigating the interactions between drugs and target proteins, thereby expediting the drug development process and enhancing the effectiveness and safety of medications [37,41,43,44,45,46]. BiFC has the capability to identify PPIs associated with various diseases, thereby offering substantial evidence for the early diagnosis and prognosis assessment of such conditions [47,48,49]. Specifically, the identification of distinct PPIs within cancer cells holds promise for the timely detection and treatment of cancer [49]. BiFC also has the potential to facilitate the creation of novel therapeutic approaches centered on protein interactions [50,51,52,53,54]. Through the manipulation of molecules that disrupt pathogenic protein interactions, the progression of diseases can be impeded; conversely, the promotion of beneficial protein interactions can enhance cellular function recovery and regeneration. In conclusion, it is anticipated that BiFC will assume a greater significance in future investigations pertaining to cell signaling pathways, disease diagnosis, and drug screening, as a result of ongoing advancements and innovations in technology. This review aims to explore the recent progress in BiFC, including technical innovations and applications in investigating PPIs in cell signaling pathways.

## 2. Overview of BiFC Development

The inception of BiFC technology can be dated back to the early 21st century, when researchers endeavored to develop a dynamic, real-time way to monitor protein interactions [55,56]. The early researchers primarily concentrated on enhancing the selection and cleavage techniques of fluorescent proteins to guarantee the stability and specificity of fluorescent signals [55,57]. Concurrently, they were consistently investigating the utilization of BiFC in various biological systems and cells to broaden its range of applications [56,57]. A ubiquitin-based split-protein sensor (USPS) was introduced in 1994, enabling the real-time monitoring of PPIs within a living cell at their endogenous locations [58]. The USPS assay is capable of detecting a transient in vivo interaction between polypeptides, as exemplified by a close proximity between Sec62p and a nascent polypeptide chain [59]. The use of fluorescent proteins in biological studies has greatly increased since the discovery of Green Fluorescent Protein (GFP) in the 1960s [60]. These important early discoveries laid a solid foundation for the emergence and development of BiFC. The BiFC technology has undergone continuous development over a span of more than two decades since its inception, with a primary emphasis on the enhancement and advancement of fluorescent proteins, photopigments, and expression vectors and an expanding range of applications (Figure 2).

### 2.1. Fluorescent Protein Used for Investigating Protein–Protein Interactions

The BiFC technology was first introduced in 2002, employing yellow fluorescent protein (YFP) as a fluorescent marker for investigating PPIs [61]. The *GFP* gene was first cloned in 1992 from cnidarian, *Aequorea victoria* [62], and GFP was then confirmed in 2000 to be characteristic of fluorescence complementation [63]. Fluorescent proteins emerged as prominent markers in these early assays owing to their high sensitivity, reliability, and versatility [55,56,57]. In 2003, the multicolor bimolecular fluorescence complementation (mcBiFC) system was developed by using eYFP and eCFP [26]. To enhance fluorescence signals in a physiological culture setting, mutations in Cyan Fluorescent Protein (CFP) and YFP were employed to create Cerulean and Venus variants, and then fluorescence resonance energy transfer with fluorescence lifetime imaging microscopy (FRET-FLIM) was integrated for terrain visualization [64,65]. The development of red fluorescent protein has expanded the scope of BiFC application. The BiFC system based on DsRed variant monomeric RFP (mRFP1-Q66T) [66] and mutant monomeric RFP (mCherry) [67] has a bright red fluorescence excitation and emission wavelength. To facilitate the direct observation of protein interactions in a physiological environment of mammals, a monomeric lumin (mLumin) BiFC system was developed utilizing the mKate variant, enabling the detection of protein interactions at 37 °C and providing a more faithful representation of protein characteristics [68]. To achieve in vivo imaging with an excitation peak above 600 nm, mNeptune was applied to BiFC [34]. The mScarlet-I-based BiFC assay was developed using a traditional β-Fos/β-Jun constitutive heterodimerization model and a rapamycin-inducible FRB/FKBP interaction system, enabling the visualization of diverse PPIs in distinct subcellular compartments with exceptional specificity and sensitivity, particularly at the physiological temperature 37 °C in live mammalian cells [69]. An orange-colored BiFC system was established using the Kusabira-Orange (KO) protein isolated from the stony coral *Fungia concinna*; as a result, the specificity and sensitivity were enhanced, thus expanding the potential applications of multicolor BiFC analysis [70]. However, the auto-fluorescent protein (AFP) fragment of the BiFC complex tends to accumulate and precipitate in vitro, making it difficult to characterize physical properties, so chimeric AFP was created by directly fusing different DNA encoding AFP fragments [71]. In summary, the localization of proteins within cells and their interactions can be ascertained through fluorescence signals, obviating the necessity for chemical alterations to the proteins.

### 2.2. Photosensitive Pigments for In Vivo Imaging

Photosensitive pigments are a type of photoreceptor in bacteria or plants that can absorb red and near-infrared light and mediate the transmission of light signals [72]. The iRFP–BiFC system, which utilizes the bacterial photosensitive pigment intrinsic red fluorescent protein (iRFP), emitting long waves, has potential applications for in vivo imaging [73]. A protein-fragment complementation assay (PCA) was introduced based on an engineered *Deinococcus radiodurans* infrared fluorescent protein IFP1.4, enabling analysis of hormone-induced signaling complexes in living yeast and mammalian cells at the nanometer resolution [74]. Based on cyanobacterial pigments, mini-red fluorescent protein 670 nanoparticles (miRFP670 nano) [75] and IFP2.0 [76] with higher signal intensity and photostability were developed. By activating photosensitive pigments under specific lighting conditions, fluorescence signals can be generated at specific time and spatial points, allowing for the observation of precise regulatory mechanisms for a certain physiological process.

### 2.3. Vector System Expressing the Fluorescent Fusion Protein

A stable and efficient expression vector of the fusion protein is critical for the BiFC experiments. Expression vectors based on the pSAT series of vectors allow for determining the interaction of a “bait” protein and multiple “prey” proteins and are thus advantageous for the implementation of multicolor BiFC in living plant cells [77]. The Gateway vector system can be applied to screen interacting proteins on a large scale, as different cDNAs can be cloned into vectors without the use of restriction enzymes and ligases [78]. However, the use of multiple vectors will lead to differences in the expression levels of fusion proteins, making it difficult to interpret the interaction results quantitatively, whereas, using the improved Gateway-compatible cloning system, multiple target fragments can be transferred into the same vector to ensure the same amount of expression [79]. Subsequently, a dual-ORF expression BiFC system (pDOE) was constructed, which ensures that the fusion protein can be introduced into transformed cells in a 1:1 concentration ratio, but the efficiency is low when multiple DNA fragments are assembled in a predetermined order [80]. Therefore, the advantages of Golden Gate and Gateway cloning methods were combined to design a set of double-compatible pGate vectors, which could assemble multiple destination sequences in a predetermined order [81]. Furthermore, the creation of an All-in-One fluorescent fusion protein (AioFFP) vector toolbox using Gibson assembly enables the simultaneous incorporation of multiple fluorescent fusion protein expression units into a single plasmid, facilitating the co-expression of multiple genes in BiFC detection [82]. In addition, it is possible to improve the vector by selecting efficient promoters, inserting linker sequences, and optimizing the fusion protein configuration to ensure the robust expression of the fusion protein within the cell.

### 2.4. The Application of BiFC

The application of BiFC in the field of life science is constantly expanding. To apply BiFC to all mammalian cells and cells that are difficult to transfect with plasmids, the gene delivery tool Multiple Bacullovirus and Mammalian (MultiBacMam) combined with BiFC allows for efficient PPI screening in cells [44]. Through the optimization of the protoplast-based transient expression system (PTES), enhanced efficiency in obtaining active protoplasts can be achieved, and when coupled with the BiFC experiment, this optimized PTES can be effectively utilized for investigating protein interactions in monocotyledonous plants [83]. Modular BiFC (MoBiFC) was developed to observe specific protein interactions in chlorophylls, and its application was extended to other organelles [84]. Organelles perform biological functions through interactions of key proteins in contact with membranes [85]. In 2023, the proximity labeling strategy (BiFCPL) was developed to study the membrane contact proteome between mitochondria and endoplasmic reticulum (MERCs) [86]. In order to further analyze the structure and protein affinity of the complex, One Pattern Analysis (OPA) was recently developed by combining FRET-FLIM and BiFC, which can be used for protein affinity analysis and provide key information such as distance relationship and orientation of PPI [87].

BiFC technology can also be used for large-scale screening of interacting proteins. Lee et al. developed an arrayed screening strategy based on protein complementation, systematically studied protein–protein interactions in living cells, and conducted large-scale screening of telomere regulators, revealing the basic molecular niche of human telomere regulation and providing valuable tools for studying mammalian signaling pathways in cells [88]. Bischof et al. established a multicolor BiFC library, which covers most transcription factors of *Drosophila* and can be used for large-scale identification and analysis of PPIs in *Drosophila* [89]. A novel experimental strategy for cell-based protein complementation assay (Cell-PCA) has been proposed, which is based on BiFC, combined with high-throughput sequencing methods, and can be used to screen proteins interacting with different decoy proteins at the whole genome level in the same live-cell context [90]. More recently, a coupled BiFC/GFP binding peptide (GBP) nanobody-based technique was utilized to unravel the function of defined dimers of transcription factors in living cells [91]. In summary, these improvements have diversified and refined the application of BiFC techniques in biological research.

## 3. Protein–Protein Interactions Revealed by BiFC in Cell Signaling Pathways

### 3.1. TOR Signaling Pathway

The target of rapamycin (TOR), a serine/threonine kinase within the phosphatidylinositol 3-kinase-related kinase family, governs a signaling network critical for cell cycle progression, protein synthesis, autophagy, and other processes necessary for cell viability and growth (Figure 3A). The TOR signaling pathway is a highly conserved molecular mechanism that regulates cell growth and metabolism in response to environmental and nutritional stimuli. Especially, this pathway regulates intracellular autophagy [92,93,94], which requires the participation of a group of autophagy-related (ATG) proteins [95,96] (Figure 3A). A reduction in TOR activity was proven to trigger the autophagy process, while phosphatidic acid (PA), a negative regulator of autophagy in animals, was found to activate the mammalian target of rapamycin complex 1 (mTORC1) [97]. In the investigation of the suppressive impact of PA on autophagy in plants, multiple assay techniques including BiFC, yeast two-hybrid (Y2H), glutathione S-transferase (GST) pull-down, Co-IP, and FRET-FLIM were used to elucidate the interaction of GAPCs (glyceraldehyde-3-phosphate dehydrogenase), PGK3 (phosphoglycerate kinase 3), and autophagy-related proteins (ATG3 and ATG6) within the endoplasmic reticulum (ER) (Figure 3A), and these complexes were notably strengthened in the presence of PA, suggesting that the interaction complex may play a role in the early regulation of autophagy in the ER [98]. When cells experience disturbances from internal and external stimuli, misfolded proteins accumulate in the ER, triggering the onset of ER autophagy [99]. Through Y2H and BiFC assays, ATG8-interacting proteins 1 and 2 (ATI1 and ATI2) were found to interact directly with AGO1 on the ER in agroinfiltrated tobacco leaves, supporting the involvement of ATI1 and ATI2 in the Argonaute (Ago)-mediated ER autophagy pathway [100]. Y2H and in vivo BiFC assays verified that ER-localized MSBP1 (Membrane Steroid Binding Protein 1) interacts with the selective autophagy cargo receptors ATI1 and ATI2 [101].

Post-translational covalent modification of histones is also involved in the regulation of autophagy [102], and mTOR can affect the transcription of histone acetyltransferase and regulate the expression of autophagy genes [103]. By using BiFC and GST pull-down techniques, it was found in *Magnaporthe oryzae* that the domain PHD (Plant Homeodomain) of Snt2 protein binds to acetylated histone H3, and the domain ELM2 binds to deacetylase Hos2 to regulate the expression of autophagy genes ATG6, 15, 16, and 22, while the expression of Snt2 is positively regulated by TOR [104] (Figure 3A). In response to cellular stress or nutrient deprivation, the sucrose non-fermenting protein-1-related protein kinase (SnRK1) inhibits TOR activity (Figure 3A), leading to the dephosphorylation of ATG13 and its subsequent binding to ATG1 to form precursor complexes, thereby initiating the process of autophagy [96,105,106]. Y2H and BiFC results showed that glycoprotein encoded by *Rice stripe mosaic cytorhabdovirus* (RSMV) interacts with SnRK1B in *Oryza sativa* to promote the kinase activity of SnRK1B on ATG6b (Figure 3A), thereby positively regulating autophagy [107]. Studies have shown that SnRK and TOR proteins play critical roles in plant sugar metabolism [108,109]. Results of Y2H and BiFC assays demonstrated that TOR in *Vitis vinifera* interacts with SnRK1.1 to regulate *SUCs* (*sucrose transporters*), *HTs* (*hexose transporters*), *HXKs* (*hexokinases*), and other glucose metabolism-related genes [110] (Figure 3A).

TOR can also phosphorylate the microtubule-associated protein tau (MAPT), leading to MAPT aggregation and inhibition of autophagy [111]. However, an anti-depressant drug, sertraline (Sert), as an autophagy inducer, can activate AMP-activated protein kinase (AMPK), thereby inhibiting the oligomerization of MAPT in MAPT-BiFC cells [43]. This negative regulation of the TOR pathway promotes the autophagic degradation of MAPT, thus suppressing tauopathy [43]. The subunits TOR1, TOR2, and Kog1 form the TOR Complex 1 (TORC1), which regulates cell growth by modulating autophagy and ribosome biogenesis processes [112,113]. A genome-wide examination of the TORC1 interactome was performed in yeast through BiFC assay, resulting in the identification of predominant BiFC signals localized at the vacuolar membrane, and a subset of these signals were found within cytoplasmic messenger ribonucleoprotein (mRNP) granules, where TORC1 modulates the activity of the translation repressor protein Scd6 via phosphorylation, thereby influencing the regulation of post-transcriptional gene expression [114].

### 3.2. PI3K/Akt Signaling Pathway

The signaling pathway PI3K/Akt (phosphatidylinositol 3′-kinase and RAC-alpha serine/threonine-protein kinase or protein kinase B) plays a crucial role in intracellular signal transduction (Figure 3B), influencing processes such as cell growth, proliferation, survival, and metabolism [115,116]. In research pertaining to this pathway, the BiFC technique is primarily utilized for the visualization and analysis of PPIs among key components. Snail and GSK-3β, integral constituents of the PI3K/Akt pathway, play a regulatory role in the epithelial–mesenchymal transition (EMT) process [117] (Figure 3B). Results of immunoprecipitation and BiFC assays showed that overexpression of ajuba LIM protein (AJUBA) can recruit tumor necrosis factor associated factor 6 (TRAF6), enhance Akt phosphorylation, activate the Akt/GSK-3β/Snail signaling pathway, and promote the EMT process, thereby enhancing the invasion and metastasis ability of HCC (Hepatocellular Carcinoma cells) [118] (Figure 3B). Epithelial membrane protein 3 (EMP3) can affect the PI3K-Akt pathway in cancer cells [119,120]. Y2H screening identified 10 previously unreported interaction partners of EMP3, eight of which were further verified through BiFC and Proximity Ligation Assay (PLA); among these candidate interactors, Flotillin-1 (FLOT1), HIV Tat-interacting protein 2 (HTATIP2), Ras-related protein 2A (RAP2A), and Proteolipid protein 2 (PLP2) exhibited strong signals in BiFC and PLA assays [121] (Figure 3B). FLOT1 and PLP2 have been identified as positive regulators of the PI3K-AKT signaling pathway, as evidenced by previous studies [122,123]. Conversely, HTATIP2 and CMTM5 (CKLF-like MARVEL transmembrane domain-containing member 5) have been shown to exert negative regulatory effects on this pathway [124,125]. Additionally, RAP2A has been found to exhibit both positive and negative regulatory capabilities in Akt signaling [126,127] (Figure 3B).

Lysosomes are known to be major organelles involved in autophagy [128,129]. Y2H and BiFC assays have substantiated the occurrence of an interaction between Akt and Phafin2 in both the cytoplasm and nucleus; nevertheless, upon treatment of cells with rapamycin or HBSS (Hank’s Balanced Salt Solution), the Akt-Phafin2 complex is augmented in the lysosome, thereby triggering autophagy [130] (Figure 3B). Subsequently, the proteins that interact with the Akt complex in the lysosomes after autophagy induction was further investigated by means of time-of-flight mass spectrometry (TOF/MS), BiFC, and immunofluorescent assays, and it was revealed that VRK2 (Vaccinia-related kinase 2) maintains Akt kinase activity by interacting with Akt in the lysosomes, thereby regulating autophagy and cell proliferation [131] (Figure 3B).

### 3.3. Wnt/β-Catenin Signaling Pathway

The primary regulator of the Wnt/β-catenin pathway is the Axin/APC/GSK3β destruction complex (DC) (Figure 3C), which allows for the degradation of cytoplasmic β-catenin in the absence of external stimulation. The stability of β-catenin protein is critical in the regulation of the Wnt signaling pathway [132] (Figure 3C). When the Wnt signaling pathway is activated, the Axin/APC/GSK3β destruction complex (DC) becomes inactive, leading to increased stability of its target protein β-catenin, which accumulates and then translocates to the nucleus [133,134,135,136] (Figure 3C). The BiFC assay demonstrated that the conformational alteration of DC is triggered by the presence of Wnt ligands, leading to the inhibition of the Axin–GSK3β interaction in *Drosophila*, thereby impeding the degradation of β-catenin [137] (Figure 3C). Subsequently, β-catenin and T-cell transcription factor (TCF) interact in the nucleus and synergistically regulate the expression of target genes MYC, CCND1, and CDKN2A to affect cell survival [138] (Figure 3C), and BiFC signals are strong in the S and G2 phases of cells [20]. The dimerization of PAC1 (pituitary adenylate cyclase-activating polypeptide) was confirmed using BiFC and the bioluminescence resonance energy transfer (BRET) assay [139], and the PAC1 dimerization is migrated from the plasma membrane to the nucleus under serum withdrawal conditions [140]. Interestingly, similar to the frizzled receptor dimer [141], the dependent basic activity of PAC1 dimerization can also activate the Wnt/β-catenin pathway, increasing the levels of β-catenin, cyclin D1, and c-myc in the pathway, and as a result, the cells show higher anti-apoptotic activity [140]. In addition, CAM-1, a ROR receptor tyrosine kinase (RTK), is an unconventional receptor of Wnt associated with neural signaling processes dependent on the Wnt pathway [142,143,144]. Through genetic and BiFC assays, it was demonstrated that presynaptic RIG-3, an immunoglobulin superfamily protein, interacts directly with the immunoglobulin domain of postsynaptic CAM-1, a nonconventional Wnt receptor, at the *Caenorhabditis elegans* neuromuscular junction (NMJ) [145] (Figure 3C). This interaction subsequently suppresses Wnt/LIN-44 signaling, thereby preserving the appropriate levels of acetylcholine receptor, AChR/ ACR-16, at the neuromuscular synapse [145] (Figure 3C).

### 3.4. NF-κB Signaling Pathway

The NF-κB (nuclear factor kappa-light-chain-enhancer of activated B cells) family in mammals consists of five closely related transcription factors: p50, p52, p65 (RelA), c-Rel, and RelB [146] (Figure 3D). NF-κB dimers formed by the two subunits p50 and p65 (RelA) are involved in regulating the expression of genes related to cell survival, immunity, and anti-apoptosis [147,148,149]. The activator protein 1 (AP-1) superfamily, as dimeric transcription factors associated with tumor development, is composed of different family protein members such as Jun and Fos [150] (Figure 3D). It was revealed by BiFC that Rel family proteins p50 and p65 in NF-κB interact with Fos and Jun and inhibit their transcriptional activities [61] (Figure 3D). Subsequently, a BiFC-based FRET (BiFC-FRET) assay was used to confirm that the trimer complex formed by the interaction between the p65 and the Fos-Jun heterodimer of AP-1 participates in the regulation of the target gene of AP-1 [65] (Figure 3D). BATF3 is a member of the ATF-like family and belongs to the AP-1 transcription factor family [151,152]. It was proven by BiFC assay that the interaction between CiBATF3 and interleukin 10 (IL-10) in *Ctenopharyngodon idella* negatively regulates the activity of NF-κB [153] (Figure 3D). GST pull-down and BiFC assays verified that p65 inhibits the activity of NF-κB pathway through interaction with the 14-3-3 proteins in the neuronal nuclei (Figure 3D), thereby protecting neurons from ischemia (I/R) injury and regulating nerve cell survival [154]. By combining BiFC and a transposon gene trap system, researchers have screened the protein Calcyclin Binding Protein (CACYBP) that interacts with p65 and found that their interaction enhances the activity of NF-κB under TNFα stimulation [155] (Figure 3D). Subcellular co-localization and BiFC assay results showed that other protein interaction complexes can also affect the activity of NF-κB. For example, the interaction between LMP1 (latent membrane protein 1) and Tmem134 (transmembrane protein 134) [156] or the interaction between the TIR (Toll-interleukin 1-resistance) domain and Toll-like receptor 3 (Toll-3) [157] can significantly activate the NF-κB pathway and enhance the immune function of the body (Figure 3D).

### 3.5. MAPK Signaling Pathway

The Mitogen-Activated Protein Kinase (MAPK) signaling pathway, which mainly consists of Mitogen-Activated Protein Kinase Kinase Kinases (MAPKKKs or MEKKs), Mitogen-Activated Protein Kinase Kinases (MAPKKs or MKKs or MEKs), and Mitogen-Activated Protein Kinases (MAPKs or MPKs), regulates a variety of cellular activities including proliferation, differentiation, survival, and death [158] (Figure 3E). This pathway is crucial for regulating immune response mechanisms [159]. Previous studies have shown that MEKK1 and MPK4 are negative regulators of innate immune response in plants, and their deletion can induce constitutive expression of pathogenesis-related genes [160,161], while the MEKK1 mutant can cause programmed cell death [161]. The BiFC assay demonstrated that MEKK1 interacts with MKK1 and MKK2 on the plasma membrane and that the interaction signals of MPK4 with MKK1 and MKK2 appear in the plasma membrane and nucleus [162] (Figure 3E). These results suggest that MEKK1, MKK1/MKK2, and MPK4 can form a kinase cascade to negatively regulate the innate immune response of plants and prevent immune hyperplasia [162]. In mammalian cells, a subfamily of 10 dual-specificity (Thr/Tyr) MAPK phosphatases (MKPs) have the capability to either recognize, bind, and dephosphorylate a singular class of MAP kinase or to modulate multiple MAPK pathways [163]. Genetic analyses and a BiFC assay were conducted to explore the role of MKP in regulating oxidative stress and pathogen defense responses, confirming that the MAPK phosphatase 2 (MKP2) and MPK3/MPK6 interaction occurs in both the cytoplasm and nucleus of *Arabidopsis* [164] (Figure 3E). In the scenario of fungal infection in *Arabidopsis*, the MKP2 and MPK3/MPK6 interaction can significantly reduce programmed cell death, referred to as the hypersensitive response (HR) [164], which is similar to the result of inhibiting MAPKs in plants [159]. Excessive accumulation of H_2_O_2_ can also trigger the HR in plants [165]. Evidence from Y2H and BiFC suggested that MPK31 in cotton *Gossypium hirsutum* (*G. hirsutum*) regulates the production of ROS and HR-like cell death through interaction with the H_2_O_2_-producing protein RBOHB [166]. Apart from the HR mediated by resistance proteins, plants can detect pathogens by surface-localized pattern recognition receptors (PRRs) via the recognition of carbohydrate-containing molecules such as fungal chitin, bacterial peptidoglycans, and extracellular ATP, which also activate multiple immune-related pathways, including the MAPK cascade [167]. It was revealed in rice (*Oryza sativa* L.) based on the BiFC assay that RLCK185 (receptor-like cytoplasmic kinase) regulates chitin-induced MAPK activation through interaction with MAPKKK11 and MAPKKK18 at the plasma membrane, and that MAPKKK18 interacts with MKK4, an upstream MAPKK component of MPK3/6, thus forming the MAPK cascade consisting of MAPKKK18–MKK4–MPK3/6 [168].

The MAPK cascade pathway is associated with the response to abiotic stress [169]. MAPKKK plays an important role as the largest gene family in the MAPK cascade [170]. To explore the physiological functions of MAPKKK in biotic and abiotic stress responses in the oilseed crop canola (*Brassica napus* L.), 15 interaction pairs between 28 MAPKKK proteins and 8 MAPKK proteins were identified via the Y2H assay and further validated through BiFC analysis [171]. The extensive interactions of MKK with MAP3K and MPK in *G. hirsutum* were revealed by Y2H and BiFC experiments, followed by VIGS (virus-induced gene silencing) assays to confirm the involvement of the MAP3K14-MKK11-MPK31 pathway in the drought stress response of cotton [172]. The subcellular localization and BiFC assays revealed that MAPK3 interacts with the proteins in the cold response pathway, ICE41, ICE87, and CBFIVd-D9 in the nucleus or in the plasma membrane of wheat (*Triticum aestivum* L.); this interaction is crucial for the negative regulation of plant cold tolerance, as MAPK3 mediates the phosphorylation and subsequent degradation of ICE and CBF proteins [173].

As an emerging biological technique, BiFC provides a powerful tool for deciphering cellular signaling pathways. The application of BiFC in cell signaling pathway research enables real-time and intuitive observation of PPIs, providing much convenience for pathway research. A large number of studies have successfully revealed the molecular mechanisms of various cellular signaling pathways using the BiFC technique. By using BiFC to gain a deeper understanding of molecular interactions, the regulatory mechanisms of signaling pathways can be revealed, providing key insights into cellular physiological processes, tissue development, and disease occurrence. In addition, this technology is expected to help identify new signaling pathway members and key regulatory factors, providing potential targets for the development of new drugs and treatment methods.

## 4. Protein—Protein Interactions Demonstrated by BiFC in Plant Hormone Signaling Pathways

In addition to being applied in the study of classic signaling pathways, BiFC has also been widely utilized in the investigation of plant hormones. The determination of the molecular interactions in the plant hormone pathways aids in elucidating the molecular mechanisms underlying plant growth, development, and stress responses, thereby laying the groundwork for the development of high-yield and stress-resistant plant varieties.

### 4.1. Ethylene Signaling Pathway

Ethylene accelerates fruit ripening, organ senescence and abscission, and promotes the differentiation of plant sexual organs [174]. The ethylene signaling pathway involves multiple families of transcription factors [175,176] (Figure 4A). Results of Y2H, GST pull-down, and BiFC assays confirmed the interaction between Ein3-binding F-box protein (MaEBF1) and MaNAC67-like protein, key components of the ethylene signaling pathway in Fenjiao banana (*Musa* ABB Pisang Awak), and their interaction further activates the promoters of starch degradation-related genes *MaBAM6* and *MaSEX4* (Figure 4A), thus facilitating fruit softening and ripening [177]. Yeast split-ubiquitin assays and BiFC studies indicated that *Arabidopsis* CPR5 (the constitutive expressor of pathogenesis-related genes 5) directly interacts with the ETR1 receptor in regulating ethylene signal transduction [178,179] (Figure 4A). BiFC and luciferase complementation imaging (LCI) assays confirmed that CPR5 in Melon (*Cucumis melo* L.) regulates the bisexual flower phenotype through interaction with ETR1 [180] (Figure 4A). The ethylene-responsive factor (ERF) belongs to the APETALA 2/ethylene response factor (AP2/ERF) superfamily and is a core component of the ethylene signaling pathway [181]. Multiple techniques including Y2H, BiFC, GST pull-down, and luciferase complementation imaging (LCI) were jointly utilized to discover that chrysanthemum CmERF3 inhibits the expression of flowering integrators (FTL1) by interacting with B-Box (BBX) family member CmBBX8 (Figure 4A), thereby maintaining the vegetative growth of *Chrysanthemum morifolium* and preventing premature flowering [182]. It was found in wheat (*Triticum aestivum* L.) through GST pull-down, Co-IP, Y2H, and BiFC assays that TabHLH094 (a basic helix–loop–helix transcription factor) and TaMYC8 (a negative regulator of cadmium-responsive ethylene signaling) form an interactive complex, reducing their ability to bind to the ERF6 promoter and thereby inhibiting the activities of 1-aminocyclopropane-1-carboxylate oxidase (ACO) and 1-aminocyclopropane-1-carboxylic acid synthase (ACS), two key rate-limiting enzymes in the process of plant ethylene biosynthesis [183] (Figure 4A). Results of Y2H, BiFC, and LCI assays showed that the protein ERF38 interacts with MYB113, enhancing the role of MYB113 on the promoters of ACS1 (Figure 4A), ultimately boosting transcriptional efficiency in eggplant (*Solanum melongena* L.) [184]. Ethylene can also regulate plant responses to abiotic stresses such as drought [185,186]. When plants experience heat stress, there is an elevation in the ethylene content present in their leaves [187]. Evidence from BiFC and Y2H assays of *Lilium longiflorum* showed that ERF110 interacts with the heat stress transcription factor (HsfA2) in the nucleus to regulate the expression of heat stress response (HSR) genes and improve heat tolerance [188] (Figure 4A). In the investigation of the functional mechanism of DREB1 in soybean (*Glycine max* L.), it was demonstrated through Y2H and BiFC experiments that two ERF-like transcription factors ERF008 and ERF106 interact with DREB1 (Figure 4A), thereby promoting the activation of drought-resistant genes *COMT*, *TDC*, and *SANT* [189]. In summary, the BiFC technique, either alone or in combination with other technologies, enables researchers to visualize PPIs and the subcellular localization of key components in the ethylene signaling pathway.

### 4.2. Auxin Signaling Pathway

The growth hormone auxin mainly includes indole-3-acetic acid (IAA), indole-3-butyric acid (IBA), naphthoic acid (NAA), etc. Auxin plays a crucial role in various physiological processes in plants, such as promoting growth and development, enhancing stress tolerance, bolstering disease resistance, and improving resistance to herbicides [190,191]. The auxin pathway relies on two core regulatory factors, indole-3-acetic acid (IAA) protein and auxin response factors (ARFs) [192,193] (Figure 4B). It was confirmed by BiFC that ARF23 and ARF29 interact with IAA28 in the nucleus of *Populus trichocarpa* (*P. trichocarpa*) to participate in the IAA signal transduction process [194] (Figure 4B). IAA recruits TOPLESS (TPL) corepressors to inhibit the transcription of ARF auxin-responsive genes [195]. Y2H and BiFC assays demonstrated in *Chrysanthemum morifolium* that CmTPL1-1, a product of a Chrysanthemum TPL/TPR family gene, interacts with CmWOX4, CmLBD38, and CmLBD36 in modulating the auxin signaling pathway to regulate the development of root bud apical meristem and lateral organs [196] (Figure 4B). ARF5 interacts with odor-related auxin-responsive IAA4 and IAA6 in *Hedychium coronarium* to regulate the transcriptional activity of terpene synthase 3 (TPS3) in synthesizing volatile compounds related to floral fragrance β-ocimene, which was verified by Y2H and BiFC assays [197] (Figure 4B). When faced with pathogen infection, indole-3-acetic acid-amido synthetase (IAAS) induces the accumulation of conjugated auxin IAA-Asp, promoting the proliferation of pathogens [198]. It was confirmed through Y2H and BiFC that IAAS interacts with the Nia-pro protein encoded by Potato virus Y (PVY), inducing the expression of IAAS, while downregulating auxin-responsive genes, *Auxin response factor 1* (ARF1), *Auxin response factor 3* (ARF3), and *small auxin upregulated RNA 3* (SAUR3) [199] (Figure 4B). Auxin enhances plant resistance to salt stress by inducing expansions and controlling cell wall plasticity [200]. It was proven through a BiFC assay that *Chenopodium quinoa* α-expansin 50 (CqEXPA50) interacts with the auxin pathway genes ARF, IAA, Gretchen Hagen 3 (GH3), and SAUR, resulting in the accumulation of photosynthetic pigments under salt stress [201] (Figure 4B). Y2H and BiFC evidence showed that the interaction between HLH85 and phosphate transporter chamber PHF1 in Sweet sorghum (*Sorghum bicolor* L.) disrupts the transport and accumulation of phosphorus (Pi) under salt stress and inhibits the expression of auxin pathway genes *PIN3* and *SAUR50* (Figure 4B), thereby reducing plant salt tolerance [202]. Collectively, by visualizing the interactions between auxin receptors and downstream signaling components, the BiFC technique has provided insights into the spatiotemporal dynamics and regulatory mechanisms of auxin signaling.

### 4.3. GA Signaling Pathway

Gibberellin (GA) signaling regulates various plant processes such as seed germination, stem elongation, root growth, flowering, and bud dormancy release in perennial woody plants [203,204,205,206,207,208]. The precise regulation of GA metabolism and signaling is crucial for plant growth and adaptation to the environment [209]. The GA signal is recognized by its nuclear receptor, GA INSENSITIVE DWARF1 (GID1), which initiates the GA signaling pathway by facilitating the interaction between GID1 and DELLA, a repressor of GA signaling [210]. GA regulates plant leaf bud dormancy and plant defense response through the GA-GID1-DELLA regulatory module, with DELLA proteins serving as key switches in GA signal transduction [211,212,213]. It was demonstrated through Y2H, BiFC, and GST pull-down assays that the C-terminal domain of PsF-box1 interacts with the DELLA member PsRGL1 in tobacco (*Nicotiana benthamiana* L.) leaves (Figure 4B), leading to the ubiquitination-dependent degradation of PsRGL1 and facilitating the release of tree peony bud dormancy [214]. The DELLA family members RGA1 and RGL1 interact with the hormone signaling regulator KNOX1 in Rape (*Brassica campestris* L.) (Figure 4B), as confirmed by Y2H and BiFC assays, facilitating bud differentiation and bolting via the GA pathway [215]. Flowering-promoting factors (FPFs) of Mango (*Mangifera indica* L.) were revealed through BiFC assays to interact with several DELLA proteins in positive regulation of both flowering promotion and root growth enhancement in response to GA treatment [216] (Figure 4B). The interaction between the auxin response factor and DELLA influences the GA and auxin signaling pathways, thereby modulating fruit development in tomato (*Solanum lycopersicum* L.) [217]. The overexpression of the atypical bHLH transcription factor SlPRE5 in tomatoes results in elevated levels of GA, and SlPRE5 interacts with bHLH proteins SlAIF1, SlAIF2, and SlPAR1 (Figure 4B), as demonstrated through Y2H and BiFC assays, to govern plant morphology and leaf chlorophyll accumulation, consequently impacting the process of photosynthesis [218]. Subsequently, it was elucidated through Y2H and BiFC assays that the interaction between an atypical basic bHLH transcription factor SlPRE3 and other bHLH proteins SlAIF1/SlAIF2/SlPAR1/SlIBH1 influences cell expansion and modulates lateral root growth associated with GA signaling [219] (Figure 4B). In conclusion, a growing body of research suggests that BiFC, either independently or in conjunction with other methodologies like Y2H and GST pull-down, offers a foundational comprehension of gene functionality in the GA signaling pathway. The BiFC assay will play a pivotal role in elucidating the complex interactions between GA and other hormonal signaling pathways, ultimately leading to the improvement of crop production and abiotic stress tolerance in plants.

### 4.4. ABA Signaling Pathway

The plant hormone abscisic acid (ABA) plays a pivotal role in regulating various physiological processes, particularly in response to abiotic stresses such as drought, salinity, and cold [220,221,222,223]. ABA and GA modulate plant growth and development through an antagonistic interplay [224]. The ABA signaling pathway is complex and involves numerous proteins that interact to transduce the hormonal signal into cellular responses (Figure 4D). In the context of the ABA signaling pathway, the BiFC assay has been used to decipher the interactions between key players. The bZIP transcription factor ABF1 was found to be phosphorylated by the SnRK2 (sucrose non-fermenting 1-related protein kinase 2) family member SAPK8 under exogenous ABA induction, and the interaction between ABF1 and SAPK8 was validated through Y2H, GST pull-down, BiFC, and kinase assays [225] (Figure 4B). Further investigation demonstrated that ABF1 interacts with the flowering negative regulatory factor FIE2 to recruit PRC2-mediated H3K27me3 modification to the target sites for inhibition, thereby delaying the flowering process of *Oryza sativa* [225]. In addition, SnRK2.6, a member of the SnRK2 family, activates the ABA signaling pathway and helps plants adapt to drought stress by controlling stomatal closure [226]. The results of Y2H, BiFC, and FLC (Firefly luciferase complementation imaging assay) experiments suggested that protein phosphatase type 2C group A (PP2AC) interacts with a photosynthetic phosphatase activator (PTPA) (Figure 4B), thereby exerting a negative regulatory effect on SnRK2.6 activity, ultimately resulting in stomatal opening in apple leaves [227]; their interaction was further verified by GST pull-down and MST (microscale thermophoresis) assays [228]. Furthermore, the results of Y2H and BiFC experiments indicated that the ABA receptor Pyrabastin Resistance 1-Like (PYL) interacts with PP2CA proteins in the presence of ABA signaling to establish a regulatory network that triggers stomatal closure in cotton *Gossypium hirsutum* L. during periods of drought [229]. Additional BiFC data also supported the notion that the interaction among PYL/PP2CA/SnRK2 proteins constitutes an ABA signal transduction module (Figure 4B), which plays a crucial role in mediating plant responses to drought stress [230,231]. Currently, there are reports indicating that members of the HD Zip transcription factor family, specifically ATHB-6 and HDZ5-6A, are involved in activating the ABA pathway in maize *Zea mays* L. and wheat *Triticum aestivum* L. to respond to drought stress [187,232]. Subsequently, it was confirmed through BiFC and GST pull-down experiments that Zmhdz9 interacts with ZmWRKY120 and ZmTCP9 (Figure 4B), promoting the expression of the key enzyme gene NCED1 in ABA biosynthesis, thereby enhancing drought resistance in *Zea mays* L. [233]. Taken together, the utilization of the BiFC assay in elucidating the ABA signaling pathway has been revolutionary, confirming established interactions and uncovering new components and controllers, thereby enhancing our understanding of this pivotal hormonal pathway.

In summary, the use of the BiFC technique to study protein–protein interactions in plant hormone pathways is of great significance for a deeper understanding of plant growth and development regulation, for revealing the molecular mechanisms by which plants respond to biotic and abiotic stresses, and also for developing new stress resistance technologies and optimizing agricultural production. In the future, the continuous application and progress of this technology will provide more scientific support and technological means for cultivating stress-tolerant varieties, promoting the development of precision agriculture and sustainable agriculture.

## 5. Optimization and Improvement of BiFC in Deciphering Protein–Protein Interactions

Despite the numerous advantages of the BiFC technique for visualizing PPIs, observing living cells in real-time, and analyzing subcellular localization, there are also limitations, including false-positive signals, limited spatial resolution, and weak fluorescence intensity, which necessitate ongoing optimization and improvement efforts (Figure 5A).

### 5.1. Reduction of False-Positive Signal

False positives may arise due to nonspecific interactions of fluorescent fragments that self-assemble into a full fluorescent protein. A micro-tagging system, Tripartite Split-GFP, which is comprised of GFP10, GFP11, and detector GFP1-9 tags, minimizes protein interference and aggregation, and displays reduced background signals in mammalian cells, largely due to its concise labeling system [234] (Figure 5B). Mutations at certain amino acid residues located in the C-terminal fragment of Superfolder GFP (sfGFP), such as R219, D220, I229, and T230, have been demonstrated to hinder the self-assembly ability of sfGFP, resulting in a decrease in false-positive signals in negative controls [235] (Figure 5B). Additionally, segmentation at residue 210 of monomeric Venus (mVenus) can effectively decrease self-assembly, eliminate background signal, and precisely detect transformed cells in conjunction with the Golgi localization marker mTurquoise2 (mTq2) [80]. By incorporating an optical marker with one of the fragmented fluorescent proteins and utilizing it as a reference signal, the detection of non-specific PPIs can be prevented through the quantification of the fluorescence ratio between the BiFC signal and the reference signal; this method is referred to as BAC-BiFC (background assessable and correctable-bimolecular fluorescence complementation) [236] (Figure 5C). In BiFC experiments, the irreversible complementary binding of the N-terminal and C-terminal fragments of fluorescent proteins may result in artefactual readouts due to weak and transient interactions. Therefore, the inclusion of positive and negative controls is essential to assess false-positive signals and ensure the accuracy and reliability of experimental outcomes [61,64]. Cells co-transfected with the expression vectors pBiFC-bJunVN173 and pBiFC-bFosVC155 may serve as positive controls to assess the functionality of the experimental system. Conversely, cells transfected with plasmids solely encoding fluorescent protein fragments, plasmids encoding one target protein fused to fluorescent protein fragments, or plasmids encoding a non-interacting mutated partner can be utilized as negative controls.

### 5.2. Improvement of Spatial Resolution

The conventional BiFC resolution is limited by light diffraction, making it inappropriate for detecting PPI at the nanoscale. When utilized in conjunction with Photoactivated Localization Microscopy (PALM) [237] and Stochastic Optical Fluctuation Imaging (SOFI) [238], the enhanced BiFC technique can be employed for the precise localization of PPI at the nanoscale level. Nevertheless, BiFC-PALM is limited to observing interacting proteins in fixed cells [237,239], whereas BiFC-SOFI necessitates imaging analysis for obtaining super-resolution images, rendering it unsuitable for high-resolution real-time PPI studies in live cells [238]. Real-time monitoring of PPI within living cells was achieved at the nanoscale resolution using Reversible Saturable Optical Fluorescence Transition (RESOLFT) nanoscopy [240]. When traditional Fluorescent Protein Indicators are transformed into photo-transformable Fluorescent Protein-based Indicators (ptFP), such as PAmCherry1 [237], mEos3.2 [239], mIrisFP [241], and rsEGFP2 [240], the identification of their cleavage sites can be determined by analyzing their individual protein structures. This approach involves the integration of super-resolution microscopy (SRM) techniques (BiFC-PALM, BiFC-SOFI, and BiFC-RESOLFT) to enable super-resolution imaging of protein interactions [242]. Furthermore, the proper manipulation of experimental materials, such as the deliberate creation of gaps between the lower epidermis and muscle layer tissues of plant leaves, has been shown to mitigate the potential interference of the muscle layer and upper epidermis on imaging quality, ultimately enhancing resolution [243].

### 5.3. Enhancement of Fluorescence Signal

Traditional BiFC methods are hindered by low signal intensity and instability, thereby restricting their capacity to identify interactions between proteins with weak affinity. The GGGSGGG-linker sequence was employed for the purpose of connecting mRFP fragments with target proteins, enhancing the flexibility of the fusion protein interaction process; this optimization has been shown to have a notable impact on signal strength [244] (Figure 5B). The improved Tripartite sfGFP can significantly enhance the fluorescence signal by capturing the interacting protein complex with the anti-GFP(1-9) nanobody (VHHr) [245]. The incorporation of Luciferase Bioluminescence Technology-High Bioluminescence Tag (LgBiT-HiBiT) onto the GFP1-9 and GFP11 fragments results in the conversion of the sfGFP Tripartite system into a novel Bipartite system, thereby substantially improving the interaction signal and signal-to-noise ratio of BiFC detection [246] (Figure 5B). The tandem near-infrared BiFC system (tBiFC) was developed by linking two fragments of IFP2.0 (Improved monomeric near-infrared phytochrome 2.0) [76] (Figure 5C) or utilizing miRFP670nano as the signal module [75], enhancing the intensity and sensitivity of the BiFC signal while exhibiting superior optical stability. Organic dyes exhibit superior brightness and light stability compared to fluorescent proteins and can be conjugated to target proteins through self-labeling [247,248]. Consequently, the Tagged Bimolecular Fluorescence Complementation (TagBiFC) system, which utilizes the Haloalkane Dehalogenase Tag (HaloTag), offers enhanced signal intensity for visualizing PPIs in live cell imaging [249] (Figure 5D). Although monomeric near-infrared (NIR) fluorescent proteins (FPs) have a longer wavelength compared to infrared FP (IFP) and possess attributes such as low cytotoxicity and suitability for deep penetration imaging, they exhibit relatively low brightness when excited by conventional lasers [250]. The monomeric IFP 663m (mIFP663) was utilized as a fluorophore to optimize excitation at 633 nm, exhibiting superior brightness, stability, and compatibility with subcellular localization in the PPI analysis of viable cells [251]. Moreover, tagged blue fluorescent protein 2 (TagBFP2) exhibits superior brightness and a superior signal-to-noise ratio in BiFC detection when compared to conventional fluorescent proteins, enabling the acquisition of high-quality images without the requirement of costly high-end equipment [15].

## 6. Concluding Remarks and Future Perspectives

The cell signaling pathway is an important process of information transfer within and between cells, which regulates cell growth, development, and function. Protein–protein interactions are one of the core mechanisms by which cell signaling pathways function. BiFC provides a way to visualize and quantify protein interactions for cell biology research. The application of BiFC in cell signal transduction research is summarized as follows: (1) The detection of protein interactions: by directly observing fluorescence signals, the interaction between proteins can be detected in real-time and dynamically, so the BiFC assay allows us to understand the regulatory mechanisms of cell signal transduction pathways. (2) Uncovering the composition of protein complexes: by combining with other protein-labeling techniques, such as immunolabeling or genetic labeling, BiFC can be used to identify the composition of protein complexes involved in specific signal transduction events. (3) Revealing the subcellular localization of protein interactions: luminescence of fluorescent proteins can provide subcellular localization information of protein interactions, which is critical for understanding the spatial and temporal regulation of specific signal transduction events within cells. (4) Drug screening and functional research: BiFC can be used to screen compounds or small molecules that can affect specific protein interactions, further providing candidate molecules for drug discovery. In addition, BiFC can also be used to study the effects of protein interactions on cell functions, such as proliferation, differentiation, and apoptosis.

The BiFC technique has a broad application prospect in the research of cell signaling pathways, and its future development trend is concentrated in the following aspects: (1) Improving signal intensity and sensitivity: enhancing the fluorescence signal intensity and sensitivity of BiFC will enable the detection and quantitative analysis of proteins with low expression levels or weak interactions, providing more accurate insights into protein interaction events. (2) Combining with multi-channel and multi-labeling techniques: by introducing fluorescent proteins of different colors or using fluorescent dyes for multi-channel labeling, multiple interaction events within the same cell can be detected simultaneously, enabling a more comprehensive understanding of the complexity of gene regulatory networks. (3) High-resolution imaging and 3D visualization: With the advancement of microscopy technology, more attention will be paid to high-resolution imaging and 3D visualization. The combination of super-resolution microscopy and imaging techniques will allow for the observation and quantitative analysis of dynamic changes in protein interactions at the subcellular level [252]. (4) Combining with high-throughput omics technology: by integrating large-scale protein interaction data and expression data, a more accurate and complete gene regulatory network model is constructed. (5) Utilizing molecular-docking and machine-learning methodologies for the prediction of PPI networks [253] and subsequently corroborating these predictions through the application of BiFC to validate the interactions of the relevant proteins. For example, AlphaFold3 exhibits exceptional capability in accurately predicting the high-precision structural interactions of various biological molecules [254].

The current BiFC technique, including in vivo applications, necessitates the use of plasmids for the expression of exogenous proteins to detect protein interactions. Thus, the challenge of observing endogenous protein interactions in vivo remains a significant issue to be addressed. Miyakura et al. [155] fused the N-terminal of Kusabira-Green (mKG) with the decoy protein and then used PiggyBac transposon to insert the sequence of the remaining mKG into the genome, randomly fusing endogenous genes and combining RACE experiments to achieve the screening of endogenous interacting proteins. This method provides a new way to observe the interaction of endogenous proteins, but the expression of the decoy protein requires the introduction of foreign plasmids. It may be possible to insert the expression sequence of fluorescent protein fragments into the genome at a fixed point, but the selection of insertion sites, the amount of endogenous protein expression, and the intensity of fluorescent protein need to be verified. Combining BiFC with other technologies, such as CRISPR-Cas9 gene editing [255,256], live-cell imaging, and ultra-high-resolution microscopy imaging [252], will further improve our understanding of protein interactions in cell signaling pathways.

## Figures and Tables

**Figure 1 biomolecules-14-00859-f001:**
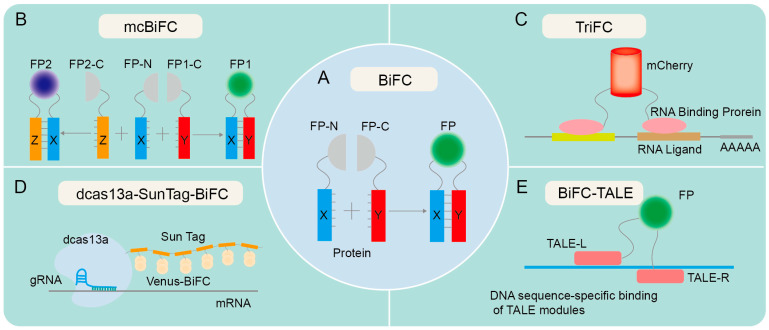
Technical principles of various forms of BiFC. (**A**) The principle of traditional BiFC. Proteins X and Y contain an N-terminal fragment and a C-terminal fragment, respectively, and are hypothesized to potentially interact with each other. The fusion of X protein and Y protein facilitates the identification of the interaction between X and Y. If X and Y proteins interact, recombination of the fluorescent protein fragments is expected to result in fluorescence emission. (**B**) The principle of mcBiFC. X protein interacts with Y and Z proteins, respectively, emitting different fluorescence due to the unique fluorescent protein fragments of Y and Z proteins. (**C**) The principle of TriFC. The N-terminal fragment of mCherry is linked to the target mRNA via an RNA-binding protein, and its C-terminus is linked to a potential RNA-binding protein. Upon interaction between the candidate RNA-binding protein and the target mRNA, the N- and C-terminal fragments of mCherry undergo recombination, resulting in the production of a red fluorescent protein (RFP) signal. (**D**) The principle of dcas13a-SunTag-BiFC. The fusion of dCas13a with SunTag tags facilitates the recruitment of fluorescent protein fragments, enabling RNA imaging via the targeted binding of guide RNA (gRNA). (**E**) The principle of BiFC-TALE. Transcription activator-like effector (TALE) can bind to specific DNA sequences and aggregate fluorescent protein fragments for imaging genomic loci.

**Figure 2 biomolecules-14-00859-f002:**
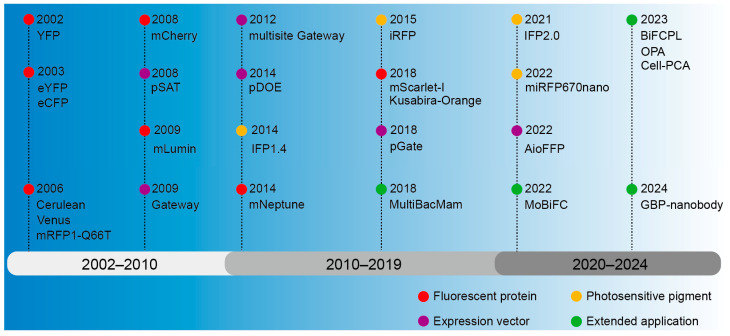
The development history overview of BiFC.

**Figure 3 biomolecules-14-00859-f003:**
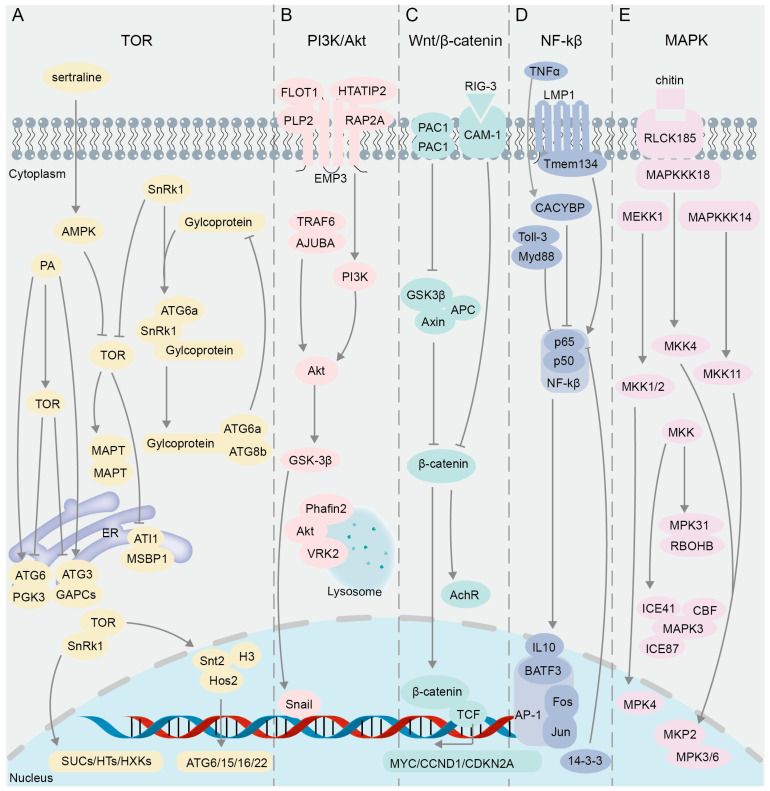
The interacting proteins in the classical cell signaling pathway validated using BiFC. (**A**) TOR signaling pathway. (**B**) PI3K/Akt signaling pathway. (**C**) Wnt/β-catenin signaling pathway. (**D**) NF-κB signaling pathway. (**E**) MAPK signaling pathway.

**Figure 4 biomolecules-14-00859-f004:**
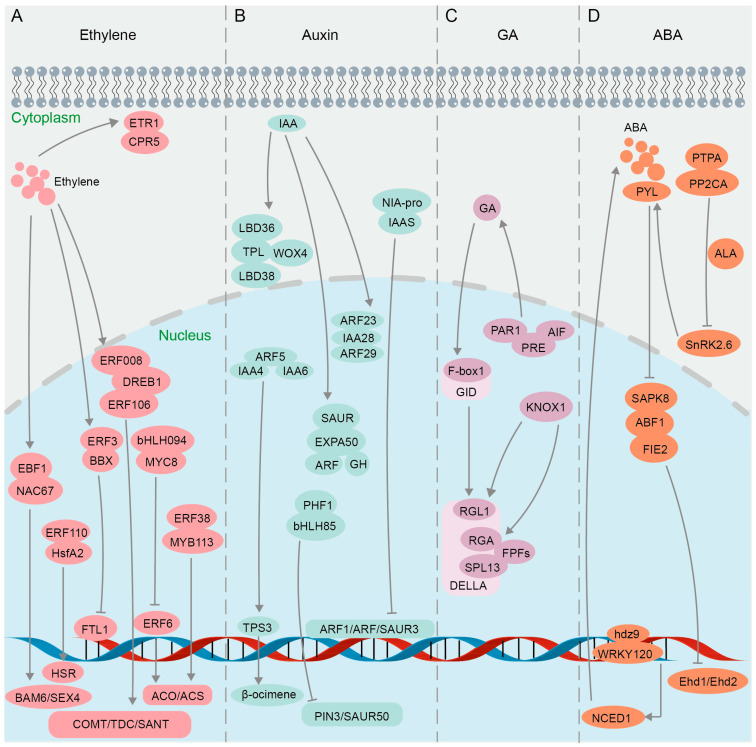
Protein interactions in plant hormone pathways confirmed using BiFC. (**A**) Ethylene signaling pathway. (**B**) Auxin signaling pathway. (**C**) GA signaling pathway. (**D**) ABA signaling pathway.

**Figure 5 biomolecules-14-00859-f005:**
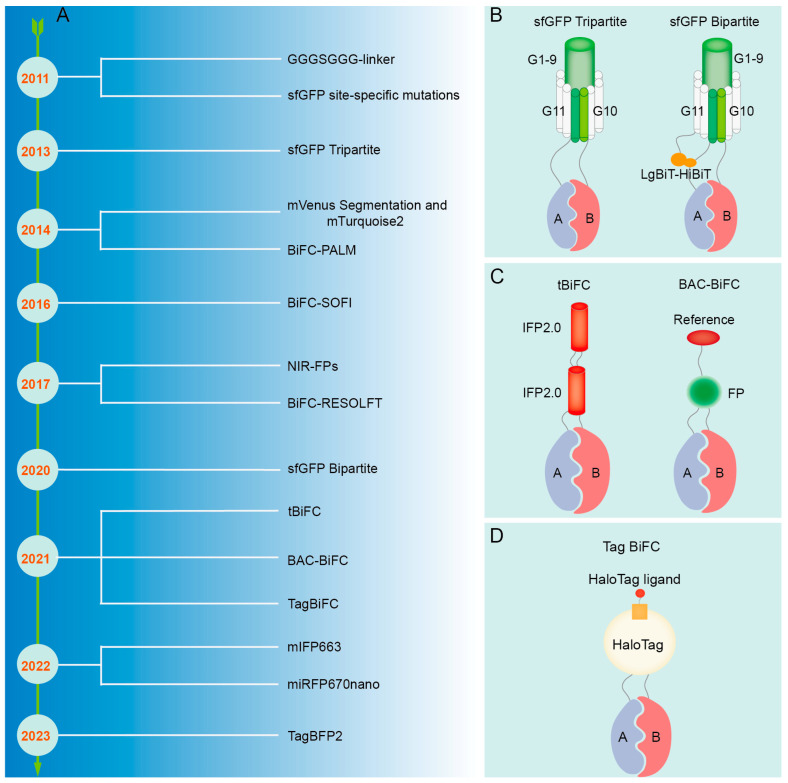
The refinement of the BiFC technique. (**A**) Improvement process of BiFC technique. (**B**) Schematic diagram of sfGFP. (**C**) Schematic diagram of tBiFC and BAC-BiFC. (**D**) Schematic diagram of Tag BiFC and sfGFP Bipartite.

## Data Availability

No data were generated for the research described in this article.

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
