# Peer review of "Comprehensive Review on Bimolecular Fluorescence Complementation and Its Application in Deciphering Protein–Protein Interactions in Cell Signaling Pathways"

_biomolecules, 2024, doi:10.3390/biom14070859_

Round 1

Reviewer 1 Report

Comments and Suggestions for Authors

The review by Houming Ren et al. summarizes and critically analyzes recent advances in bimolecular fluorescence complementation, focusing on its application for the detection and analysis of interactions relevant in cell and plant hormone signaling. The fundamentals of the technique and its evolution are first described, addressing key aspects for the design of BiFC assays such as the fluorescent proteins and vector systems for their expression available. After a brief summary of general applications of the methodology for the detection of all kinds of protein interactions in cells, the authors thoroughly review studies in which BiFC was instrumental to unravel interactions concerning several cell signaling cascades of utmost relevance and a similar number of plant hormone signaling pathways. The article also examines potential avenues for improvement of the sensitivity and resolution of the technique.

The manuscript is very well structured and the information nicely presented. The figures and schemes are pertinent and very helpful to illustrate key aspects. Overall, this is a nice piece of work that should be useful for a variety of readers interested in the technique.

Minor issue

There is a typo in line 205

Author Response

The review by Houming Ren et al. summarizes and critically analyzes recent advances in bimolecular fluorescence complementation, focusing on its application for the detection and analysis of interactions relevant in cell and plant hormone signaling. The fundamentals of the technique and its evolution are first described, addressing key aspects for the design of BiFC assays such as the fluorescent proteins and vector systems for their expression available. After a brief summary of general applications of the methodology for the detection of all kinds of protein interactions in cells, the authors thoroughly review studies in which BiFC was instrumental to unravel interactions concerning several cell signaling cascades of utmost relevance and a similar number of plant hormone signaling pathways. The article also examines potential avenues for improvement of the sensitivity and resolution of the technique. The manuscript is very well structured and the information nicely presented. The figures and schemes are pertinent and very helpful to illustrate key aspects. Overall, this is a nice piece of work that should be useful for a variety of readers interested in the technique. Response: We are very grateful to you for your valuable comments on our manuscript.

Reviewer 2 Report

Comments and Suggestions for Authors

The review of Ren et al. provides an updated overview of the technical advancements of BiFC, starting with various applications and concluding on optimization and improvements of the method in term of specificity, resolution and fluorescence intensity. The core of the review is centered on the use of BiFC in deciphering protein-protein interactions (PPIs) in different animal and plant signaling pathways.

This review is nicely written, timely appropriate and comprehensively retraced the evolution of BiFC tools over the last 10 years.

Several points should be mentioned to make this review even more complete for readers:

-       Describe the type of negative controls that should be used for any BiFC assay (need of controlling the expression levels, competition with cold partner, use of a mutated partner that could not interact) in a dedicated part of the chapter 2.

-   Mention/highlight that BiFC is irreversible, which may lead to artefactual readouts due to the abnormal stability of the BiFC protein complex over time

-       The authors did not add references once stating the combination of BiFC and CRISPR-CAS9 technology or Super-resolution microscopy and BiFC knowing that several papers discussed these techniques (Wilson, E.L., Yu, Y., Leal, N.S. et al. (2024). https://doi.org/10.1038/s41419-024-06568-y, Hong Y et al. 2018 doi: 10.1186/s13059-018-1413-5, Valli et al. 2021 https://doi.org/10.1016/j.jbc.2021.100791)

-       In the part 2.4 (Applications of BiFC) authors should mention work related to

  - large scale interaction screens with BiFC (Ding Z; et al., https://doi.org/10.1073/pnas.060691710; Lee OH et al., https://doi.org/10.1074/mcp.M110.001628; Cooper SE et al., https://doi.org/10.1080/15384101.2015.1053667; Jia et al., doi: 10.3390/cells12010200. 

  - in vivo analysis of PPI networks (Bischof et al., doi: 10.7554/eLife.38853.)

  - a recent work coupling BiFC and a specific nanobody for studying the activity of transcription factors (Miyake and McDermott, doi: 10.1093/nar/gkae548)

-   In the part 5.3 (enhancement of Fluorescence signal), authors mentioned the Tripartite BiFC with sfGFP fragments. In fact the original paper cites by the authors illustrates the advantage of Tripartite BiFC over BiFC at the level of specificity, not stronger fluorescent signal (it is rather the opposite). This work should therefore be mentioned in the part 2.1. A more recent paper showed that Tripartite BiFC could be used for high resolution analysis of PPI when coupled to a nanobody. This paper should be mentioned in the part 5.3 (Castillo et al., https://doi.org/10.1016/j.ejcb.2023.151355).

Author Response

Comment 1:

The review of Ren et al. provides an updated overview of the technical advancements of BiFC, starting with various applications and concluding on optimization and improvements of the method in term of specificity, resolution and fluorescence intensity. The core of the review is centered on the use of BiFC in deciphering protein-protein interactions (PPIs) in different animal and plant signaling pathways.

This review is nicely written, timely appropriate and comprehensively retraced the evolution of BiFC tools over the last 10 years.

Response: We are very grateful to you for your comments on our manuscript.

Comment 2:

Several points should be mentioned to make this review even more complete for readers:

(1) Describe the type of negative controls that should be used for any BiFC assay (need of controlling the expression levels, competition with cold partner, use of a mutated partner that could not interact) in a dedicated part of the chapter 2.

Response: We are grateful to the reviewer for your valuable advice. Considering the logic, fluency and completeness of each part of the article, the statement about negative BiFC controls was not listed in a dedicated part of the chapter 2 but added to 5.1. Reduction of False Positive Signal. Please see lines 632-641.

(2) Mention/highlight that BiFC is irreversible, which may lead to artefactual readouts due to the abnormal stability of the BiFC protein complex over time

Response: Thank you for your comments. The complementary binding of the N-terminal and C-terminal of fluorescent proteins is irreversible, and false positive read-outs might be generated by weak and transient interactions that can be captured using BiFC, so negative controls need to be designed in any BiFC assays to rule out false positives. Please see lines 633-635.

(3) The authors did not add references once stating the combination of BiFC and CRISPR-CAS9 technology or Super-resolution microscopy and BiFC knowing that several papers discussed these techniques (Wilson, E.L., Yu, Y., Leal, N.S. et al. (2024). https://doi.org/10.1038/s41419-024-06568-y, Hong Y et al. 2018 doi: 10.1186/s13059-018-1413-5, Valli et al. 2021 https://doi.org/10.1016/j.jbc.2021.100791)

Response: We are grateful to the reviewer for the valuable suggestions. These three referenceshave been cited. Please see [255, 256] and [252] in lines 749-752. The following information is only for checking.

[255] Wilson, E.L., Yu, Y., Leal, N.S. et al. (2024). https://doi.org/10.1038/s41419-024-06568-y.  

[256] Hong Y et al. 2018 https://doi.org/10.1186/s13059-018-1413-5.

[252] Valli et al. 2021 https://doi.org/10.1016/j.jbc.2021.100791.

(4) In the part 2.4 (Applications of BiFC) authors should mention work related to

①large scale interaction screens with BiFC (Ding Z; et al., https://doi.org/10.1073/pnas.060691710; Lee OH et al., https://doi.org/10.1074/mcp.M110.001628; Cooper SE et al., https://doi.org/10.1080/15384101.2015.1053667; Jia et al., doi: 10.3390/cells12010200.

Response: The paper (Ding Z; et al., https://doi.org/10.1073/pnas.060691710) is not available, probably this DOI is not correct.

The paper (Cooper SE et al., https://doi.org/10.1080/15384101.2015.1053667) is an example of the application of BiFC but is not related to large scale interaction screens with BiFC, so it is not cited in the revised version.

The other two papers (Lee OH et al., https://doi.org/10.1074/mcp.M110.001628 and Jia et al., https://doi.org/10.3390/cells12010200) were cited as [88] and [90] in the revised version.

② in vivo analysis of PPI networks (Bischof et al., doi: 10.7554/eLife.38853.)

Response: The paper was cited as [89] Bischof et al., doi: 10.7554/eLife.38853.

③ a recent work coupling BiFC and a specific nanobody for studying the activity of transcription factors (Miyake and McDermott, doi: 10.1093/nar/gkae548)

Response: We greatly appreciate your suggestions and we have added relevant content and cited these references. This paper was cited as [91] Miyake and McDermott, doi: 10.1093/nar/gkae548.

(4) In the part 5.3 (enhancement of Fluorescence signal), authors mentioned the Tripartite BiFC with sfGFP fragments. In fact the original paper cited by the authors illustrates the advantage of Tripartite BiFC over BiFC at the level of specificity, not stronger fluorescent signal (it is rather the opposite). This work should therefore be mentioned in the part 2.1.

Response: Thank you for your suggestion. This statement about the Tripartite BiFC with sfGFP fragments has been moved from part 5.3 to part 5.1. Please see lines 618-622. The original paper was cited as [234] (Cabantous S, Nguyen HB, Pedelacq JD, Koraichi F, Chaudhary A, Ganguly K, Lockard MA, Favre G, Terwilliger TC, Waldo GS: A new protein-protein interaction sensor based on tripartite split-GFP association. Scientific reports 2013, 3:2854.).

(5) A more recent paper showed that Tripartite BiFC could be used for high resolution analysis of PPI when coupled to a nanobody. This paper should be mentioned in the part 5.3 (Castillo et al., https://doi.org/10.1016/j.ejcb.2023.151355).

Response: Thank you for your suggestion. This paper has been cited in the revised manuscript. Please see lines 669-670. The improved Tripartite sfGFP can significantly enhance the fluorescence signal by capturing the interacting protein complex with anti-GFP (1-9) nanobody (VHHr) [245] (Castillo S, Gence R, Pagan D, Koraïchi F, Bouchenot C, Pons BJ, Boëlle B, Olichon A, Lajoie-Mazenc I, Favre G et al: Visualizing the subcellular localization of RHOB-GTP and GTPase-Effector complexes using a split-GFP/nanobody labelling assay. European Journal of Cell Biology 2023, 102(4):151355.). 

Reviewer 3 Report

Comments and Suggestions for Authors

It would be good to give examples of some of the most successful fluorophore pairs with their chemical structure, spectral properties and ways of using them.

Author Response

Comments and Suggestions for Authors

It would be good to give examples of some of the most successful fluorophore pairs with their chemical structure, spectral properties and ways of using them.

Responses: Thank you for your valuable suggestion. We have cited some papers on the fluorophore pairs, the relevant statements could be found in this manuscript but were not all detailed in a dedicated part in the revised version. Please see lines 149-158, 678-680, 681-683, 686-688.